# Diagnostic Value of Cytomegalovirus IgM Antibodies at Birth in PCR-Confirmed Congenital Cytomegalovirus Infection

**DOI:** 10.3390/ijms20133239

**Published:** 2019-07-01

**Authors:** Shohei Ohyama, Kazumichi Fujioka, Sachiyo Fukushima, Shinya Abe, Mariko Ashina, Toshihiko Ikuta, Kosuke Nishida, Hisayuki Matsumoto, Yuji Nakamachi, Kenji Tanimura, Hideto Yamada, Kazumoto Iijima

**Affiliations:** 1Department of Pediatrics, Kobe University Graduate School of Medicine, Kobe 650-0017, Japan; 2Department of Clinical Laboratory, Kobe University Hospital, Kobe 650-0017, Japan; 3Department of Obstetrics and Gynecology, Kobe University Graduate School of Medicine, Kobe 650-0017, Japan

**Keywords:** cytomegalovirus IgM antibody, congenital cytomegalovirus infection, quantitative real-time PCR

## Abstract

Although cytomegalovirus (CMV) DNA detection in urine is the standard method for diagnosing congenital cytomegalovirus infection (CCMVI), polymerase chain reaction (PCR) is not comprehensively available. Currently, the efficacy of CMV-specific IgM (CMV-IgM) and CMV-specific IgG (CMV-IgG) detection remains unclear. To determine the sensitivity and specificity of CMV-specific antibodies at birth, we investigated CMV-IgM and CMV-IgG titers in CCMVI cases and non-CCMVI controls, with confirmed diagnoses by urine quantitative real-time PCR within 3 weeks after birth. We included 174 infants with suspected CCMVI in whom serological testing was performed within the first 2 weeks after birth during 2012–2018. We classified the participants into a CCMVI group (*n* = 32) and non-CCMVI group (*n* = 142) based on their urine PCR results. The CMV-IgM-positive rate was 27/32 (84.4%) in the CCMVI group, compared with 1/142 (0.7%) in the non-CCMVI group (*p* < 0.0001). The positive CMV-IgG rates were 32/32 (100%) in the CCMVI group and 141/142 (99.3%) in the non-CCMVI group. The positive predictive value for CMV-IgM was high at 96.4% (27/28). This value may be sufficient for clinical use, especially in settings with limited resources where PCR is unavailable. However, CCMVI screening by CMV-IgM alone appears insufficient because of the considerable number of false-negative cases.

## 1. Introduction

Cytomegalovirus (CMV) is a virus that causes mother-to-child infections, and congenital CMV infection (CCMVI) can result in non-hereditary hearing impairment and severe developmental disorders [1,2]. We previously reported that abnormal fetal ultrasonographic findings and positive CMV results by PCR in maternal cervical mucus were independent risk factors for predicting fetal CMV infection in pregnant women with positive CMV-specific IgM (CMV-IgM) antibody test results [3]. In recent years, detection of CMV DNA in urine within 3 weeks after birth has become the standard for the diagnosis of CCMVI [1]. However, the PCR technique is not comprehensively available in general obstetric clinics and is thus clinically inconvenient.

Although a previous study investigated the efficacy of CMV-IgM for the diagnosis of CCMVI [4], no reports have clarified the efficacy of CMV-IgM and CMV-specific IgG (CMV-IgG) for the diagnosis of CCMVI in infants with PCR-confirmed CCMVI diagnoses as positive controls and PCR-confirmed non-CCMVI infants as negative controls.

In this study, we investigated CMV-IgM and CMV-IgG titers in neonatal serum from PCR-confirmed CCMVI cases and non-CCMVI controls to determine the sensitivity and specificity of these antibodies for the diagnosis of CCMVI among newborns with mothers suspected of having CMV infection.

## 2. Results

### 2.1. Patient Characteristics

We performed univariate analyses in 174 infants. Patient characteristics are shown in Table 1. Regarding symptoms at birth, thrombocytopenia (*p* < 0.001), microcephaly (*p* = 0.04), hearing dysfunction (*p* < 0.001), brain computed tomography (CT) abnormality (*p* < 0.001), eye complications (*p* < 0.001), and small for gestational age (SGA; *p* < 0.001) were significantly lower in the non-CCMVI group than in the CCMVI group.

### 2.2. Laboratory Results in Congenitally Infected and Uninfected Neonates

The antibody positivity rates are shown in Table 2. The CMV-IgM-positive rates were 27/32 (84.4%) in the CCMVI group and 1/142 (0.7%) in the non-CCMVI group, with a significant difference (*p* < 0.001). The CMV-IgG-positive rates were 32/32 (100%) in the CCMVI group and 141/142 (99.3%) in the non-CCMVI group, with no significant difference.

## 3. Discussion

In this study, we found that the detection of CMV-IgM in neonatal serum had a sensitivity of 84.4% and a specificity of 99.3% for the diagnosis of CCMVI in our cohort.

Revello et al. [4] reported that CMV-IgM had a sensitivity and specificity of 70.7% and 100%, respectively, for CCMVI diagnosis in their cohort. Although their specificity was similar to that in the present study (100% vs. 99.3%), their sensitivity was much lower (70.7% vs. 84.4%). There were two important differences between the two studies: Revello et al. used urine virus isolation as a CCMVI diagnostic method, and had a much smaller number of non-CCMVI controls (*n* = 34 vs. *n* = 142 in the present study). Because PCR has become the standard diagnostic method, the diagnostic accuracy for CCMVI may be superior in the present study compared with the previous study. Regarding specificity, we examined CMV-IgM in 142 non-CCMVI cases, which is four times the number tested by Revello et al. Thus, we could confirm the extremely low incidence of false-negative cases. Nelson et al. [5] reported that CMV-IgM had a sensitivity and specificity of 25% (5/20) and 100% (32/32), respectively. Although their high specificity was similar to the present finding, they adopted virus culture for CCMVI diagnosis, similar to Revello et al., and thus their method differed from the quantitative real-time RT-PCR (qRT-PCR) method used in the present study, which has substantially higher sensitivity. Bilavsky et al. [6] reported that CMV-IgG had a sensitivity of 40.7% in a cohort study on 199 patients with CCMVI. Although the number of patients in their study was higher than that in the present study, the weaknesses in their study included: (1) the diagnosis of CCMVI was based on more than a single method (viral culture or PCR) and (2) CMV-IgM measurements were performed qualitatively by comparing absorbances between participant specimens and cut-off specimens. These differences may contribute to the lower sensitivity of CMV-IgM in their study. In the present study, we used a quantitative enzyme immunoassay (EIA) method to measure CMV-IgM in both the CCMVI and non-CCMVI groups diagnosed by qRT-PCR, which allowed us to clarify the highly accurate clinical efficacy of CMV-IgM.

The limitation of the present study is that we could not collect data on the timing of the initial maternal CMV infections during pregnancy. The reason is that 25% (8/32) of the CCMVI cases were outborns, and thus the type and timing of testing among mothers were inconsistent. Moreover, the maternal symptoms caused by primary CMV infection are nonspecific, and it is difficult to discriminate CMV from other viral infections in the clinical setting. As a future study, evaluation of CMV-specific antibody titers in mothers over time from early pregnancy may make it possible to estimate the timing of primary infection according to the time of CMV-IgG-positive seroconversion.

The present study revealed that the detection of CMV-IgM at birth had a sensitivity of 84.4% and a specificity of 99.3% for the diagnosis of CCMVI. In addition, the positive predictive value of CMV-IgM was high at 96.4% (27/28). This value may be sufficient for clinical use, especially in settings with limited resources where PCR is unavailable. However, CCMVI screening using CMV-IgM alone appears insufficient because of the presence of a considerable number of false-negative cases.

## 4. Materials and Methods

### 4.1. Study Design and Patients

As part of a prospective cohort study for universal CMV screening at Kobe University Hospital [7,8,9], we examined CMV-specific antibodies, CMV antigenemia, and CMV DNA in urine samples collected on filter paper, and measured urine viral loads by qRT-PCR in neonates suspected of having CCMVI. The study was conducted under approval from the Ethics Committee of Kobe University Graduate School of Medicine (#923, 1 September 2009), and parental written informed consent was received for all participants.

To determine the diagnostic value of CMV-specific antibodies at birth, we included 174 infants suspected of having CCMVI based on their maternal history or postnatal course during a 7 year study period (January 2012 to December 2018), and in whom CMV-IgM and CMV-IgG testing was performed within the first 2 weeks after birth, based on a previously published protocol [6]. Diagnosis of CCMVI was confirmed by positive qRT-PCR results for urine taken within 3 weeks after birth [8,10,11,12]. Urine viral load was measured by qRT-PCR, based on our previous report [12]. Based on the presence or absence of CCMVI, we classified the participants into two groups: the CCMVI group and the non-CCMVI group. The positive rates for CMV-IgM and CMV-IgG were compared between the groups.

Clinical characteristics, including gestational age, birth weight, outborn status, sex, Apgar scores, neonatal asphyxia, thrombocytopenia, liver dysfunction, microcephaly, hearing dysfunction, abnormalities on brain CT, eye complications, SGA, and symptomatic CCMVI, were retrospectively collected from patient charts. Laboratory data including CMV antigenemia and initial CMV viral load in urine were also collected.

### 4.2. Definitions of Neonatal Morbidities

Neonatal asphyxia was defined as an Apgar score of ≤6, which is a scoring system used to assess the overall condition of newborns after birth [13]. Outborns were defined as infants born at locations other than Kobe University Hospital. Microcephaly was defined as head circumference less than −1.5 the standard deviation of the mean value for Japanese newborns of the same gestational age [14]. SGA was defined as birth weight less than the 10th percentile of the mean value for Japanese newborns of the same gestational age [14]. Liver dysfunction was defined as serum aspartate aminotransferase level >100 U/L, and thrombocytopenia was defined as platelet count <1 × 10^5^/μL. Eye complications were defined as CMV-associated retinopathy, such as chorioretinitis, diagnosed by a pediatric ophthalmologist. Hearing dysfunction was diagnosed based on auditory brainstem response abnormalities using a Neuropack S1 (Nihon Kohden Co., Tokyo, Japan), whether unilateral or bilateral, including absent wave V to 40 dB or 50 dB in infants at a postconceptional age of 37 weeks or 34–36 weeks, respectively. Brain CT abnormalities included intracranial calcifications, ventricular dilation, white matter abnormalities, and cortical dysplasia [10]. CCMVI was considered “symptomatic” if at least one of the following symptoms was present: thrombocytopenia, liver dysfunction, microcephaly, hearing loss, head image abnormality, ophthalmic abnormality, and SGA [15].

### 4.3. CMV-Specific Antibody Tests and CMV Antigenemia

CMV-IgM and CMV-IgG antibody titers were measured by EIA using specific detection kits for CMV-IgM and CMV-IgG (Denka Seiken Co., Tokyo, Japan). A CMV-IgM-positive result was defined as an antibody index value ≥0.8, and a CMV-IgG-positive result was defined as an EIA value ≥2.0, according to the manufacturer’s instructions. CMV antigenemia was measured by an EIA-based plasma C7-HRP assay (CMV antigen test; TEIJIN TFB, Tokyo, Japan) targeting the lower matrix protein pp65, as described previously [16]. CMV antigenemia was considered positive for at least 1 positive cell per 50,000 leukocytes [17].

### 4.4. Statistical Analysis

Data are expressed as median (range) or number (percentage). Univariate analyses were performed by the Mann–Whitney nonparametric rank test, chi-squared test, or Fisher’s exact test, as appropriate, to compare data between the two groups. All analyses were carried out with Statcel Ver. 3 software (OMS Publishing Inc., Saitama, Japan). Differences were deemed statistically significant at *p* < 0.05.

## 5. Conclusions

The positive predictive value of CMV-IgM was high enough for clinical use, especially in resource-limited settings. However, CCMVI screening using CMV-IgM alone appears insufficient because of the presence of a considerable number of false-negative cases.

## Figures and Tables

**Table 1 ijms-20-03239-t001:** Clinical characteristics of infants with and without CCMVI diagnoses confirmed by PCR.

Clinical Characteristics	CCMVI Group (*n* = 32)	non-CCMVI Group (*n* = 142)
Gestational age, weeks	37 (24–40)	**38 (28–42) ****
Birth weight, g	2344 (715–3274)	**2955 (904–3898) ****
Outborns	8/32 (25)	**0/142 (0) ****
Male	18/32 (56)	79/142 (56)
Apgar score at 1 min	8 (1–9)	**8 (1–10) ****
Apgar score at 5 min	9 (5–10)	**9 (2–10) ****
Neonatal asphyxia	8/32 (25)	**5/142 (4) ****
Thrombocytopenia	9/32 (28)	**1/142 (1) ****
Liver dysfunction	4/32 (13)	6/142 (4)
Microcephaly	4/32 (13)	**5/142 (4) ***
Hearing dysfunction	14/31 (45)	**0/142 (0) ****
Brain CT abnormality	19/31 (61)	**2/137 (1) ****
Eye complications	3/31 (10)	**0/142 (0) ****
Small for gestational age	11/32 (34)	**8/142 (6) ****
Symptomatic CCMVI	25/32 (78)	-

* *p* < 0.05 versus CCMVI group. ** *p* < 0.01 versus CCMVI group. Data are shown as median (range) or number (percentage). CCMVI, congenital cytomegalovirus infection; CT, computed tomography; PCR, polymerase chain reaction.

**Table 2 ijms-20-03239-t002:** Results for CMV-specific antibody titer, CMV antigenemia, and initial CMV viral load in urine by PCR in congenitally infected and uninfected neonates.

Laboratory Testing for CMV	CCMVI Group (*n* = 32)	non-CCMVI Group (*n* = 142)
Timing of initial neonatal serologic testing, day	1 (0–14)	**0 (0–11) ***
Timing of urine qRT-PCR, day	1 (0–20)	**0 (0–6) ***
CMV IgM antibody positive	27/32 (84)	**1/142 (1) ****
CMV IgM antibody titer	3.74 (0–12.69)	**0 (0–0.99) ****
CMV IgG antibody positive	32/32 (100)	141/142 (99)
CMV IgG antibody titer	16.7 (5.7–77.3)	**9.6 (0–58.4) ****
CMV antigenemia positive	22/32 (69)	**0/28 (0) ****
CMV antigen-positive cells per 50,000 cells	3 (0–71)	-
Initial CMV viral load in urine	3.75 × 10^7^ (1.0 × 10^4^–3.1 × 10^9^)	-

* *p* < 0.05 versus CCMVI group. ** *p* < 0.01 versus CCMVI group. Data are shown as median (range) or number (percentage). CCMVI, congenital cytomegalovirus infection; CMV, cytomegalovirus; IgM, immunoglobulin M; IgG, immunoglobulin G; qRT-PCR, quantitative real-time PCR.

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
