# Peer review of "Diagnostic Value of Cytomegalovirus IgM Antibodies at Birth in PCR-Confirmed Congenital Cytomegalovirus Infection"

_ijms, 2019, doi:10.3390/ijms20133239_

Round 1

Reviewer 1 Report

The manuscript is much improved.

Author Response

We thank you for your warm comments to our paper.  We will send the editor the certificate of English language editing from Edanz.

Reviewer 2 Report

Ohyama et. al. have resubmitted a manuscript wherein they describe the value of IgM antibodies for the diagnosis of congenital CMV infection. As IgM serology in congenitally infected children has been previously examined at least 3 times (4-6), the novelty of this report is low. Nonetheless, the succinctly presented results of this study may be of some interest to the community.

1)      As written, the Results section largely repeats information that is found in either Table 1 or Table 2. Please consider revising to better describe the study and emphasize the most important pieces of information.

2)      Changes to the materials and methods section should be considered:

a.       4.3 “Definitions of study groups” should be included in 4.1

b.       4.2 “Definition of symptomatic CCMVI” section describes all clinical definitions and the title should be revised accordingly.

c.       Lines 160-166 describing CMV EIA and antigenemia assays should receive their own header.

3)      Section 5—conclusions is inconsistent with lines 118-122.

Author Response

We thank the reviewer for the thoughtful review and helpful suggestions.

Ohyama et. al. have resubmitted a manuscript wherein they describe the value of IgM antibodies for the diagnosis of congenital CMV infection. As IgM serology in congenitally infected children has been previously examined at least 3 times (4-6), the novelty of this report is low. Nonetheless, the succinctly presented results of this study may be of some interest to the community.

Response: We thank the reviewer for the warm comment.

Point 1: As written, the Results section largely repeats information that is found in either Table 1 or Table 2. Please consider revising to better describe the study and emphasize the most important pieces of information.

Response 1: We agree with reviewer’s comment. We deleted the sentences and revised the Result section to emphasize the most important pieces of information.

P2, Line56-60.

We performed univariate analyses in 174 infants. Patient characteristics are shown in Table 1. Regarding symptoms at birth, thrombocytopenia (p<0.001), microcephaly (p=0.04), hearing dysfunction(p<0.001), brain computed tomography (CT) abnormality(p<0.001), eye complications (p<0.001), and small for gestational age (SGA; p<0.001)were significantly lower in the non-CCMVI group than in the CCMVI group.

P3, Line78-81.

The antibody positivity rates are shown in Table 2. The CMV-IgM-positive rates were 27/32 (84.4%) in the CCMVI group and 1/142 (0.7%) in the non-CCMVI group, with a significant difference (p<0.001). The CMV-IgG-positive rates were 32/32 (100%) in the CCMVI group and 141/142 (99.3%) in the non-CCMVI group, with no significant difference.

Point 2:Changes to the materials and methods section should be considered:

a.     4.3 “Definitions of study groups” should be included in 4.1

b.       4.2 “Definition of symptomatic CCMVI” section describes all clinical definitions and the title should be revised accordingly.

c.       Lines 160-166 describing CMV EIA and antigenemia assays should receive their own header.

Response 2: We have followed the reviewer’s advice and revised the materials and methods section.

P4, Line143- P5, Line146

Based on the presence or absence of CCMVI, we classified the participants into two groups: CCMVI group and non-CCMVI group. The positive rates for CMV-IgM and CMV-IgG were compared between the groups. 

P5, Line153-156.

4.2. Definitions of neonatal morbidities

Neonatal asphyxia was defined as ≤6 for the Apgar score, a scoring system used to assess the overall condition of newborns after birth[13].Outborns were defined as infants born at locations other than Kobe University Hospital.

P5, Line171-178.

4.3. CMV-specific antibody tests and CMV antigenemia.

CMV-IgM and CMV-IgG antibody titers were measured by EIA using specific detection kits for CMV-IgM and CMV-IgG (Denka Seiken Co., Tokyo, Japan). A CMV-IgM-positive result was defined as antibody index value ≥0.8, and a CMV-IgG-positive result was defined as EIA value ≥2.0, according to the manufacturer’s instructions. CMV antigenemia was measured by an EIA-based plasma C7-HRP assay (CMV antigen test; TEIJIN TFB, Tokyo, Japan), targeting the lower matrix protein pp65, as described previously [16]. CMV antigenemia was considered positive for at least 1 positive cell per 50,000 leukocytes [17].

Point 3: Section 5—conclusions is inconsistent with lines 118-122.

Response 3: We deeply agree with reviewer’s comment. We revised the conclusion section to be consistent with the last paragraphs of the discussion section. 

P5, Line187- P6, Line 198.

5. Conclusions

The positive predictivevalue of CMV-IgM was high enough for clinical use, especially in resource-limited settings. However, CCMVI screening using CMV-IgM alone appears insufficient because of the presence of a considerable number of false-negative cases. 

We would like to thank the reviewer once again for his or her useful commentsconcerning our paper. We hope that the revised manuscript is suitable for publication.

This manuscript is a resubmission of an earlier submission. The following is a list of the peer review reports and author responses from that submission.

Round 1

Reviewer 1 Report

The authors should explain how it differs from their earlier paper in this journal

Tanimura K, Yamada H. Potential Biomarkers for Predicting Congenital Cytomegalovirus Infection. Int J Mol Sci. 2018 Nov 27;19(12). pii: E3760. doi: 10.3390/ijms19123760.

After reading the Introduction and Results sections, I cant see whether the antibodies were sought in cord blood, the mothers (& when) or the babies (& when).

Terms and abbreviations are not defined adequately in The Results section

Eg: Delayed vs standard CCMVI, Apgar scores, SGA, outborns

Why are total Ig titres important?

Line 104 reads

“Second, when the CMV IgM test was performed more than 2 weeks after birth, the diagnostic accuracy was decreased.”

Presumably this reflects the definition of “delayed”….which presumably will be provided later…but I am confused because the std and delayed groups had different clinical signs in Table 1.

The Discussion is repetitive -it could be halved in length.

Author Response

Response to Reviewer 1 Comments

We thank the reviewer for the thoughtful review and helpful suggestions.

Point 1: The authors should explain how it differs from their earlier paper in this journal

Tanimura K, Yamada H. Potential Biomarkers for Predicting Congenital Cytomegalovirus Infection. Int J Mol Sci. 2018 Nov 27;19(12). pii: E3760. doi: 10.3390/ijms19123760.

Response 1: Dr. Tanimura has described how to detect CCMVI high risk pregnant women from the perspective of obstetricians in his review paper.They have risk-stratified the targets by maternal serology and reported CMV IgM positive pregnant women as high risk. Although they mentioned about newborns screening of CCMVI, the CMV IgM and CMV IgG appearing in their text are all maternal. Thus, both papers differ greatly in that we have examined the relationship between neonatal CMV IgM and the presence or absence of CCMVI, regardless of maternal serology.

Point 2:After reading the Introduction and Results sections, I cant see whether the antibodies were sought in cord blood, the mothers (& when) or the babies (& when).

Response 2: We measured the antibodies in neonatal serum. We added the description in the introduction and discussion section as below. And regarding the timing of sampling, we have shown the data in the Table 2.

P2. Line50

P4. Line103 

In this study, we found two important observations. First, when using CMV IgM in neonatal serum

Table 2.

Timing of initial neonatal serologic testing, day

Point 3: Terms and abbreviations are not defined adequately in The Results section, Eg: Delayed vs standard CCMVI, Apgar scores, SGA, outborns

Response 3: According to reviewer’s suggestion, we have defined the terms and abbreviations. Especially for delayed vs standard CCMVI groups, we have revised the description for these two groups as “CCMVI-serology ≤ 2 weeks” group and “CCMVI-serology > 2 weeks” group in all sentences for further clarification. In addition, we have revised the other inadequate terminology as below.

P2 Line59,

higher incidence of outborns (Infants born at a location other than Kobe University Hospital,

P2 Line68-69, 

and small for gestational age(SGA, p<0.01) were significantly lower

P6 Line179-180, 

Neonatal asphyxia was defined as Apgar score (a scoring system to assess newborn’s overall condition after birth)≤6 [14].

P6 Line181-183, 

SGA was defined as birth weight less than 10 percentile of the mean values of Japanese newborns of the same gestational age[15].

Point 4: Why are total Ig titres important?

Response 4: We agree with the reviewer that total Ig titers are not important in this study in terms of investigating the efficacy of CMV specific antibody for the diagnosis of CCMVI. We have deleted the description about total Ig titers from the result and discussion section to reduce the total length.

Point 5: Line 104 reads

“Second, when the CMV IgM test was performed more than 2 weeks after birth, the diagnostic accuracy was decreased.” Presumably this reflects the definition of “delayed”….which presumably will be provided later…but I am confused because the std and delayed groups had different clinical signs in Table 1.

Response 5: We apologize to you for our confusing terminology. As we added in the method section, all CCMVI diagnosis was done within three weeks of birth based on positive PCR results, and only the blood draw and CMV serology was delayed in our old version of  “CCMVI-delayed” group. Thus, we changed the terminology from “CCMVI-delayed” to “CCMVI-serology > 2 weeks” as mentioned above.

Point 6:  The Discussion is repetitive -it could be halved in length.

Response 6: Thank you for your comment, we have deleted the length of the discussion section.

We would like to thank the reviewer once again for his or her useful commentsconcerning our paper. We hope that the revised manuscript is suitable for publication.

Reviewer 2 Report

In “Diagnostic value of cytomegalovirus IgM antibody in PCR-confirmed congenital cytomegalovirus infection,” Ohyama and colleagues quantified IgM levels in a cohort of children that had been tested for CCMVI using qRT-PCR diagnostics. Several previous studies have assessed CMV-IgM levels (4-6) in children with diagnosed CCMVI, though these reports have either used culture-based diagnosis or a combination of culture and PCR-based diagnostics, may have lacked a control group, or utilized qualitative rather than quantitative ELISA. While previous studies have reported poor sensitivity in CMV IgM testing, the authors report a particularly strong predictive value to CMV IgM testing when the assay is completed on samples collected before two weeks of age, suggest that the IgM assay may have some value in resource-limited settings, and their results would be reasonable justification for a larger prospective study.

1)      The CCMVI standard versus CCMVI delayed terminology is confusing. The methods state that all CCMV diagnostics occurred within three weeks of birth in all cases. Only the blood draw and CMV serology was delayed in the “CCMVI-delayed” group. Please consider an alternative description for these two groups.

2)      Though there are significant differences between the CCMVI standard and CCMVI delayed groups, the rationale for considering the groups separately is not stated. Delayed testing (after 2 weeks) seems to have been the result of the retrospective study design and patient referral.

a.       There may be some value in subdividing the delayed group and reporting data for samples collected at specific times (e.g. 2 weeks to 1 month, 1 month to 2, 2 and beyond).

b.       Were there any non-CCMVI patients who were referred to this study that were examined after 2 weeks that are not presented in this report?

3)      Please include data for the “Timing of CCMV qRT-PCR” median (range) (ideally in Table 2) if this information is available.

4)      1 CCMVI(-) newborn had a positive IgM result. Was diagnostic PCR or viral culture repeated on this individual to rule out an incorrect initial test?

5)      Line 136: Should read: “…CMV IgM test was performed more than 2 weeks after birth…”

Author Response

Response to Reviewer 2 Comments

We thank the reviewer for the thoughtful review and helpful suggestions.

In “Diagnostic value of cytomegalovirus IgM antibody in PCR-confirmed congenital cytomegalovirus infection,” Ohyama and colleagues quantified IgM levels in a cohort of children that had been tested for CCMVI using qRT-PCR diagnostics. Several previous studies have assessed CMV-IgM levels (4-6) in children with diagnosed CCMVI, though these reports have either used culture-based diagnosis or a combination of culture and PCR-based diagnostics, may have lacked a control group, or utilized qualitative rather than quantitative ELISA. While previous studies have reported poor sensitivity in CMV IgM testing, the authors report a particularly strong predictive value to CMV IgM testing when the assay is completed on samples collected before two weeks of age, suggest that the IgM assay may have some value in resource-limited settings, and their results would be reasonable justification for a larger prospective study.

Response: We thank the reviewer for the warm comment.

Point 1: The CCMVI standard versus CCMVI delayed terminology is confusing. The methods state that all CCMV diagnostics occurred within three weeks of birth in all cases. Only the blood draw and CMV serology was delayed in the “CCMVI-delayed” group. Please consider an alternative description for these two groups.

Response 1: We agree with reviewer that our terminology was confusing. We revised the description for these two groups as “CCMVI-serology ≤ 2 weeks” group and “CCMVI-serology > 2 weeks” group, respectively.

Point 2:Though there are significant differences between the CCMVI standard and CCMVI delayed groups, the rationale for considering the groups separately is not stated. Delayed testing (after 2 weeks) seems to have been the result of the retrospective study design and patient referral.

Response 2: We have followed the protocol described by Bilavsky et al (Bilavsky E. et al., Positive IgM in Congenital CMV Infection.Clin Pediatr (Phila) 2017, 56, (4), 371-375.). They performed serology testing during the first 2 weeks of life. We regarded that serological testing within 2 weeks after birth as a standard approach for CCMVI diagnosis. In addition, we also tried to examine the usefulness of CMV IgM when serological tests were performed after 2 weeks of birth, which typically happened to the infants referred from other hospital.

P6, Line204-205

Based on the presence or absence of CCMVI and the timing of CMV IgM and IgG testing whether within the first 2 weeks of life or not based on the previously published protocol[6],

a.      There may be some value in subdividing the delayed group and reporting data for samples collected at specific times (e.g. 2 weeks to 1 month, 1 month to 2, 2 and beyond).

Response 2a: We have subdivided the CCMVI-serology > 2 weeks group and added the information as supplementary table S1, S2. And we have confirmed that there was no significant difference among the subgroups except the timing of testing.

P6, Line211-213

In addition, we have categorized the CCMVI-serology > 2 weeksgroup into 3 subgroups by the timing of serology testing (15-30d, 31-60d, and 61d-, Supplementary Table S1, 2).

b.       Were there any non-CCMVI patients who were referred to this study that were examined after 2 weeks that are not presented in this report?

Response 2b: There were no non-CCMVI patients who were referred to our hospital that were examined after 2 weeks during the study periods. We have added the sentences below.

P6, Line209-211

There were no non-CCMVI patients who were referred to our hospital that were examined after 2 weeks during the study periods.

Point 3: Please include data for the “Timing of CCMV qRT-PCR” median (range) (ideally in Table 2) if this information is available.

Response 3: We have added the data for the timing of urine qRT-PCR into the Table 2. And through this checkup, we have found that 1 case in the non-CCMVI group missed the result of urine qRT-PCR (although we confirmed negative PCR result of urine samples collected on filter paper in this case), thus we have excluded this case from the study (n=143 to 142 in non-CCMVI group). In addition, we have found 7 CCMVI cases whose qRT-PCR performed more than 3 weeks after birth, thus we have changed the basis of the definitive diagnosis to the positive PCR result in urine samples collected on filter paper taken soon after birth (n=5) or positive PCR results using dried umbilical cord specimens (n=2). We believe our diagnostic accuracy for CCMVI has not declined since all cases was still diagnosed by PCR even after revision. We deeply apologize to you for our mistakes. 

Table 2

Timing of urine qRT-PCR, day

0.5 (0-20)

21.5 (10-98)

0 (0-6)

P3 Line80-81

Of the 176 infants, 34 (19.3%) were diagnosed with CCMVI and 142 (80.7%) were diagnosed as non-CCMVI by PCR.

P4 Line112

Since the PCR method has become the standard, the

P5 Line160-161

CMV DNA in urine samples collected on filter paper,

P5 Line167-171

University Hospital. Diagnosis of CCMVI was confirmed by positive PCR results for urine (n=169) or urine samples collected on filter paper (n=5) taken within 3 weeks after birth [8, 10-12]. Urine viral load was measured using qRT-PCR, based on our previous report [12]. In 2 cases where diagnosis using urine of the affected child was impossible because they have been refereed after 3 weeks of age, CCMVI was diagnosed by positive PCR results using dried umbilical cord specimens [13].

Point 4: 1 CCMVI(-) newborn had a positive IgM result. Was diagnostic PCR or viral culture repeated on this individual to rule out an incorrect initial test?

Response 4: As we added in the Materials and Methods section, we have provided universal CMV screening by using urine samples collected on filter paper, in addition to targeted urine qRT-PCR. Thus, this CCMVI negative newborn with a positive IgM result has been tested by CMV PCR twice (urine samples collected on filter paper + urine), resulting in repeated negative results. So, we believe that this case should not be an incorrect initial test.

P5, Line160-161

we examined CMV-specific antibody, CMV antigenemia, CMV DNA in urine samples collected on filter paper, and urine viral load was measured using quantitative real-time PCR (qRT-PCR)

Point 5: Line 136: Should read: “…CMV IgM test was performed more than 2 weeks after birth…”

Response 5: Sorry for our typographical error. We have revised the sentence.

P4, Line 126

We found that when the CMV IgM test was performed more than 2 weeks after birth, the diagnostic

We would like to thank the reviewer once again for his or her useful commentsconcerning our paper. We hope that the revised manuscript is suitable for publication.

Round 2

Reviewer 1 Report

This manuscript has no new techniques and the issue addressed is not completely novel. Hence it should be clear, concise and instantly informative. However it remains really hard to read. This begins with the abstract which is written to suggest that antibodies were not sought in urine! You need to avoid complex nouns (long strings of modifiers)

I am also concerned with the design. Why is 3 weeks the cut-off for detection of CMV in urine? Two children were tested later with dried blood...were others tested and found to be negative?

Why was 2 weeks adopted as a cut-off for serological testing? Was there a drop-off in detection of IgM over days of life? Are we assuming they were all infected at or before birth so the IgM declined after 2 weeks. A longitudinal study would show this and is feasible with n=20

What does the detection of CMV antigen add to the study?

All babies were CMV IgG positive. Were their mothers also positive? Were the babies breastfed?